# Post-Craniopharyngioma and Cranial Nerve-VI Palsy Update on a MS Patient with Major Depression and Concurrent Neuroimmune Conditions

**DOI:** 10.3390/brainsci9100281

**Published:** 2019-10-17

**Authors:** Navzer D. Sachinvala, Angeline Stergiou, Duane E. Haines, Armen Kocharian, Andrew Lawton

**Affiliations:** 1United States Department of Agriculture, Agricultural Research Service, USDA-ARS, New Orleans, Home 2261 Brighton Place, Harvey, LA 70058, USA; 2Department of Medicine, Fairfield Medical Center, 401 North Ewing, Lancaster, OH 43130, USA; drangelinestergiou@gmail.com; 3Department of Neurobiology and Anatomy, Wake Forest School of Medicine, 1 Medical Center Boulevard, Winston-Salem, NC 27157, USA; dhaines@wakehealth.edu; 4Department of Neurology, Wake Forest School of Medicine, 1 Medical Center Boulevard, Winston-Salem, NC 27157, USA; 5Department of Neurobiology and Anatomy, The University of Mississippi Medical Center, Jackson, MS 39216, USA; 6Department of Radiology, Houston Methodist Hospital, 6565 Fannin Street, Houston, TX 77030, USA; akocharian@houstonmethodist.org; 7Department of Ophthalmology, Ochsner Hospital, 1514 Jefferson Highway, New Orleans, 70121, USA; andrew.lawton@ochsner.org

**Keywords:** major depression, multiple sclerosis, bupropion, *S-*adenosylmethionine, vitamin D3, yoga, craniopharyngioma, fractionated stereotactic radiation treatments, sphenoid sinusitis, cranial nerve-VI palsy

## Abstract

We report the case of a male multiple sclerosis (MS) patient with type 2 diabetes (T2D), asthma, major depression (MD or major depressive disorder, MDD), and other chronic conditions, after his recent difficulties with craniopharyngioma and cranial nerve-VI (CN6) palsy. In addition, we show magnetic resonance image and spectroscopy (MRI, MRS), Humphrey’s Visual Field (HVF), and retinal nerve fiber layer thickness (RNFLT) findings to explain the changes in the patient’s health, and discuss the methods that helped/help him sustain productivity and euthymia despite long-standing problems and new CNS changes.

## 1. Introduction

Earlier we discussed the case of a male multiple sclerosis (MS) patient (primary author, N.D.S.) who has many concurrent conditions including asthma, type 2 diabetes (T2D), Ehlers–Danlos syndrome (EDS), infections that take long to heal, and major depression (MD or MDD), and listed the methods he learned to be euthymic, by: Quelling pain and disability.Enduring medication side effects and noting what worked and what stopped working after a while.Subduing MD with bupropion, and supplements *S*-adenosylmethionine (SAMe) and vitamin-D3 (vit-D3).Maintaining routines for all medications, self-hypnosis, yoga, and physical exercises to stay fit and lucid.And academically studying symptoms and potential remedies for his ailments to engage in physician-assisted autoexperiments to discover solutions that provide him lasting relief [1].

Furthermore, the same methods that helped him attain and maintain euthymia with existing illnesses were successful in enabling him to be positive and productive despite new added medical diagnoses. For example, they worked: During his craniopharyngioma diagnosis and bitemporal vision loss.Through fractionated stereotactic radiation treatments (FSRT) to shrink his perisellar tumor and regain peripheral vision.And in his bout with cranial nerve-VI (CN6) palsy, diplopia, and their resolution.

In this report, we update our earlier case report with magnetic resonance imaging (MRI), magnetic resonance spectroscopy (MRS), Humphrey’s Visual Field (HVF), and retinal nerve fiber layer thickness (RNFLT) data to explain the patient’s status with regards to his MD, MS, tumor, vision, CN6 palsy, and overall health to suggest that even with new health difficulties the above methods he chanced upon continue to keep him positive and productive [1]. We write our reports to encourage readers (scientists, physicians, and patients) to discuss their own experiences in peer-reviewed forums to create insights that could empower the chronically ill to individualize their care with physician involvement and improve the quality of their lives.

## 2. Case Report 

In Reference [1], we reported the case of a male MS patient (primary author, N.D.S.), now 63, who has battled MD since childhood. His early disablements, which continue to date, include: amblyopia (left eye, OS), dyslexia, color blindness, asthma, urticaria, allergies to multiple foods, drugs, and narcotic pain medications, and susceptibility to infections that take weeks to resolve with antibiotics [1]. At 19, he was diagnosed selective immunoglobulin-M (IgM) deficient [2], and by his mid-40 s, required multiple rhinoplasties for apnea, lens replacements for premature cataracts, and uvula removal for swelling and apnea. He belongs to a closed Indo-Iranian minority, Zoroastrians (aka Parsees or Parsis), wherein enforced consanguineous relations for over a century have produced some children with neuroimmune maladies, e.g., Ehlers–Danlos syndrome [3], multiple sclerosis [4,5], cancer [6], and Alzheimer’s, cardiovascular, and Parkinson’s diseases [6,7,8]. Notwithstanding health limitations, he grew-up in and enjoys a stable family, and cherished a productive career as an accomplished scientist [9]. At 49 (2005), he developed fulminant MS that left him depressed (Beck Depression Inventory, BDI scores 30–40) [10], mobility impaired, and unable to balance and coordinate his hands. Despite intense therapies, three years later, he accepted disability retirement, and to-date is unable to coordinate his hands to manipulate small objects, play his piano and guitar, and tolerate hot environments [1]. 

Soon after MS was diagnosed, the patient received guidance from his former teachers and colleagues who were with him in graduate school in Iran (1977–1979), and to whom this paper is dedicated, to:Academically study his immune-related illnesses through university courses, textbooks, and journals.Maintain regular depression inventories.Have consistent routines for yoga and self-hypnosis, sleep, and fitness exercises.Note daily changes in serum glucose levels so that with prescribed medications (see Box 2 in Reference [1]), diet, and above routines his glucose levels are maintained between 75 and 120 mg/dL [11,12,13,14].Quell pain and associated anger with yoga and self-hypnosis, and use NSAIDs when pain becomes disabling.Judiciously use prescribed medications and note what worked and what aggravated liver enzyme levels, skin rashes, or other side reactions.And study new literature methods that could improve health, mood, and physical and cognitive functioning to engage in physician-assisted autoexperiments that afford sustained relief.

By employing the above approaches between 2005 and 2015, the patient with much effort regained several lost functions, e.g., emerged from bed and wheelchair to walking briskly with crutches. Likewise, after years of experiments with antidepressants, he discovered that bupropion augmented with *S*-adenosylmethionine (SAMe) and vitamin-D3 (vit-D3), in addition to other medications (see reference [1], Box 2), and routines keep him pain-free, alert, and productive. 

In late April 2017, he complained of heat intolerance, dizziness, falling, headaches, and blurry vision. Tests over the year revealed that he had craniopharyngioma, a benign perisellar tumor that was growing in place, pressed against his pituitary, anterior optic chiasm, surrounding tissues, and disrupted ophthalmic functions [15]. By late November, his conjoined papillary and adamantinous tumors had enlarged to ~2.3 and 0.7 mL, respectively, caused headaches and bitemporal hemianopsia (peripheral vision loss), and deteriorated R/S vision (from 20/25 and 20/40 in Sept-2016, to 20/60 and 20/100 in Nov-2017). However, his hypothalamic pituitary and adrenal, HPA, and axis hormone levels were unaffected [1,15,16,17,18].

From April 2017 to September 2018, the patient could not operate his automobile, and to-date reads text with the Kurzweil Reading Program using ≥14-point font projected on a 140 cm TV monitor, ~1 m from him [19]. 

Throughout the ordeal, he kept his BDI scores as low as possible (~mid 20 s) with daily: bupropion, SAMe, vit-D3, Yoga, self-hypnosis, exercises, regular schedule for other medications (see box 2 in Reference [1]), and visual imagery of himself scuba diving among coral and marine animals, studying, publishing, lecturing, and maintaining a positive outlook [20].

Given the patient’s conditions, asthma, MS, T2D, EDS, and susceptibility to infections, almost all neurosurgeons he consulted advised to ablate his tumors with 30 rounds of fractionated stereotactic radiation treatments (FSRTs), instead of surgery [21]. Gradually his difficulties lessened. 

Herein we explain the before and after statuses of his tumor, vision, recent bout with CN6 palsy, and diplopia (Figure 1, Figure 2 and Table 1), and conclude by describing his overall health, mood, and the challenges he must overcome. At present, challenges from old and new ailments are managed, by first becoming free of pain and disability (or lessening them), and then, engrossing self intellectually to problem-solving, so that new and long-standing maladies become puzzles to solve rather than factors that perpetuate helplessness.

## 3. Materials and Methods

### 3.1. Brain Imaging and Spectroscopy 

Brain images and spectra of the tumor were obtained by coauthor AK at the Houston Methodist Hospital, on a General Electric (GE) Healthcare (Waukesha, WI, USA), Model: MR750, 3Tesla instrument with software level DV 25.1 R03. Images were recorded without and with intravenous gadolinium contrast to show new lesions, changes in existing MS lesions with time, as well as postradiation size changes in the suprasellar cyst using the pituitary diffusion imaging protocol. Single voxel MRS data on the tumor were collected using short and long times to echo (TE = 35 ms, and TE = 144 ms, respectively) [15,16,17,18].

### 3.2. Vision Changes

Vision changes were studied annually by coauthor AL at the Ochsner Hospital, Jefferson Avenue, New Orleans. RNFL measurements were performed using Spectralis^®^ Tracking Laser, Model HRA/Spectralis with 870 nm diode laser, Heidelberg Engineering, Inc. (Franklin, MA, USA). A circular scan and a 3.4 mm circle were centered around the optic disc of each eye, and RNFLTs in microns were analyzed against data from age- and sex-matched Caucasians in the database (Table 1). Humphrey’s Visual Field data were recorded using Ziess Humphrey HF analyzer Model 750i, Meditec Inc., Dublin, CA, USA. After patient data, time of test, and R- or S-eye to be tested were recorded, the Swedish interactive threshold algorithm, SITA, was set to perform fast automated kinetic perimetry (duration ~3 min). The stimulus was white light of intensity 30 dB that illuminated an area of ~2 mm^2^ every 200 milliseconds on a chamber screen with background light intensity set at 31.5 apostilbs (ASB). The fixation monitor was gaze at blind spot, the target was central, and threshold was set at 24-2 (which is 24 degrees-temporal, 30 degrees-nasal, to record data at 54 points). Sensitivity at foveae for right and left eyes were measured at 38 and 37 decibels (dB), respectively. Right and left pupil diameters were 4.7 and 4.5 mm, respectively, and correction for both eyes was +4.75 diopter (D). Moreover, on most occasions, fixation losses for both eyes were 2/10, which suggested data were reliable. Thereafter, patient data were compared against data from age-matched Caucasian males in the HVF database (Figure 1).

## 4. Results and Discussion

### 4.1. Brain Imaging and Spectroscopy 

The patient’s MRIs before and after FSRT showed occipital lobe bending (OLB), a sign of chronic MD [1,22]. His conjoined tumor volume before FSRT was ~3 mL (Figure 1A). However, 15 months after FSRT, only the papillary tumor could be seen (V = 0.18 mL), and proton (^1^H) spectra of the papilloma, before and after FSRT, showed the same alkyl resonances of cholesterol and fatty acids at δ ~1.5 ppm and of compounds with hydroxyl groups at δ ~4 ppm, likely water and inositol. In addition, proton spectra of adamantinous tumors are undefined. Since no other perisellar tumor shows these characteristics, his CTs, MRIs, MRSs, and endocrine workup were used to tentatively diagnose craniopharyngioma without surgical biopsy. This was supported by the patient’s inability to efficiently control infections and asthma, and his difficulties with EDS, T2D, MS, and other immune problems [16,17,18].

### 4.2. Vision Changes 

At maximum tumor volume (ca 3 mL, Figure 1A, December 2017), his R/S vision was 20/70 and 20/100, he had bitemporal hemianopsia (Figure 1B,C), and his left-temporal retinal nerve fiber layer thickness (RNFLT) was 44 microns (Table 1), and the statistical average for RNFLT for men his age was 72 microns. This suggested that in December 2017, the patient’s temporal RNFLT was (1−44/72) ×100) ~39% below average for healthy Caucasian men of his age; however, RNFLTs in other sectors were within normal range at that time [23,24,25]. Statistical values of age- and sex-matched RNFLTs were from the Ziess Humphrey HF analyzer Model 750i, Meditec Inc., Dublin, CA database. At minimum tumor volume 2019 (ca 0.18 mL), his R/S vision had improved to 30/40 and 20/60, respectively; however, his RNFLT was 40 microns (Table 1), which was (1−40/72) ×100) ~ 44% below average thickness, and may have been due to inflammation of his sphenoid sinuses in April 2019 (Table 1). 

Figure 1B–D show Humphrey’s visual field (HVF) patterns for right (OD) and left (OS) eyes. The long red arrows point to blank regions (blind spots) in the temporal hemifields in Figure 1B,D. Statistical data in 1C from the HVF database enabled comparison of the patient’s retinal sensitivity to light with male subjects of his age in the HVF database. Figure 1B showed that in 2017, the patient had bitemporal hemianopsia before FSRTs (because dark patterns appeared mostly in the R and L temporal sides of the grids). It is well known that when tumors grow just above the pituitary and press against the anterior optic chiasm, patients experience bitemporal vision loss. For an excellent tutorial and discussion on visual deficits caused by lesions in visual pathways, the reader is advised to consult Reference [25], pages 264–267. After FSRT, Figure 1D data showed that the patient had central scotomas (blue arrows, points along the central meridians of retinae where vision was impaired). HVF pattern deviation maps are used to compare a patient’s retinal sensitivity to 30 decibel light with statistical information from healthy age- and sex-matched subjects in the HVF database. HVF tests evaluate retinal sensitivity at 54 points, in superior (S) and inferior (I) quadrants of temporal (T) and nasal (N) hemifields. Vision losses shown in HVF maps are graded in terms of <5%, <2%, <1%, and <0.5%. So, a four-dot rectangular pattern with a <5% designation in Figure 1C, when present in the patient’s maps (Figure 1B,D), indicated that in that region of his retina, 95% of age-matched men had better vision than the patient. Likewise, the blackened rectangle with a <0.5% designation, when present in his maps Figure 1B,D, showed that in that retinal region, 99.5% of age-matched men had better vision than him. A patient’s HVF pattern deviation maps ascribe her/his visual field defects in terms of anopsias and scotomas. Anopsias describe vision loss in quarter (quadrantanopia) or half fields (hemianopsias, Figure 1B), and scotomas describe vision loss at individual test points (Figure 1D, blue arrows). Since the highest collection of HVF patterns shown in Figure 1C was present in both temporal S and I quadrants in Figure 1B, his data showed that he had bitemporal hemianopsia before radiation treatments. Fifteen months after FSRT, as his tumor shrank, his visual acuity improved, and he showed central scotomas in both eyes (Figure 1D), which in time could resolve as inflammation lessens [15,16,17,18,26].

The RNFL is a layer of retinal nerve fibers (axons) that eventually form the optic nerve, which enters the eye at the optic cup (blind spot) and proceeds to the brain. Table 1 compares the patient’s RNFLTs in microns (µm) in S, I, N, and T regions (or sectors) of both retinas, with statistical data on healthy Caucasian men of his age in the RNFLT database. RNFLTs are measured using optical coherence tomography (OCT). Superior and inferior sectors of the eye are further divided into temporal superior and inferior (TS and TI), and nasal superior and inferior (NS and NI) sectors. The letter “G” in Table 1 indicates global or foveal RNFLT. Foveae are pits in retinal surfaces where visual acuity is highest due to the highest concentration of cone- and rod-shaped neuron cell bodies therein. Patient’s RNFLTs that were within normal limits in comparison with statistical values from matched subjects are shown in black font. Those in **blue bold** font were borderline thicknesses, and those in **red bold** font indicated below normal retinal thicknesses (calculation shown above). Overall, his data showed that this patient could gradually lose central and temporal vision in his left eye, and needs to address the problem with a retina specialist [23,24,25,27].

Losses in RNFL thicknesses (Table 1) and visual fields (Figure 1) are known to occur with age in both women and men and are exacerbated in aging patients with chronic inflammatory comorbidities (obesity, diabetes, asthma, multiple sclerosis, and/or brain tumors). Furthermore, the likelihood of craniopharyngioma recurrence is also higher in obese diabetics. Therefore, it is imperative for this patient to curb inflammation, reduce weight, keep daily serum glucose values within normal ranges, and annually monitor his ophthalmic, metabolic, and CNS (brain and spine) changes to preclude recurrence of tumor and/or further retinal damage [15,16,17,18,23,24,25,27]. 

### 4.3. Bout with CN6 Palsy

In November 2018, the patient experienced sinus pain and intracranial pressure that did not abate with yoga, self-hypnosis, saline rinses, and naproxen. He experienced diplopia and orthostatic difficulties when fixating on vertical edges of walls to balance when standing-up, which to the patient felt like the two same vertical edges were side by side (binocular double vision). In addition, he had a slight convergent right eye squint, that is, his right eye was turning slightly inwards towards his nose. Furthermore, while driving, he saw phantom traffic lane lines obliquely cutting into his lane from the right. He stopped driving and managed diplopia with a right eye patch. Soon diplopia worsened, interfered with reading and daily functions; and by month’s end (December), he was prescribed a 10-diopter base out Fresnel lens prism over his right eye (by coauthor, A.L.) after his misalignment was measured by alternate cover testing with hand-held prisms [28]. In addition, his neuro-ophthalmologist advised that the patient’s diplopia might be due to right CN6 palsy, triggered by a sinus infection that might be causing his pain and intracranial pressure. Alternatively, given the patient’s medical history, it could be due to his chronic diabetes and/or MS, and that such palsies are almost always temporary. At New Years, as nasal lavage thickened, appeared dark, and became more painful, the patient was prescribed amoxicillin to reduce infection. By late January, he was symptom free [22,26,28,29]. In March 2019, sinus pain, pressure, and diplopia returned, and at that time he was undergoing his postradiation therapy follow-up. His transaxial and coronal MRIs in addition to his tumor’s status showed that his left sphenoid sinus was more inflamed than the right (Figure 2A and B, yellow triangles). Moreover, while MRIs showed no congestion and no legion(s) in the right pons CN6 region (Figure 2C, orange arrow), the effects of intracranial pressure on CN6 could not be ruled out. Since temporary diplopia is known to be due to ipsi- as well as contralateral CN6 palsies induced by sphenoid sinusitis and increased intracranial pressure, it was conjectured that his symptoms will subside as inflammation diminishes. Later that month, his pain and diplopia were resolved with saline rinses, naproxen, and yoga. CN3 (oculomotor nerve) palsy was ruled out because the patient’s right eyelid was not droopy (no ptosis), the right pupil was not dilated, and the eye was not shifted temporally. CN4 (trochlear nerve) palsy was ruled out because diplopia did not worsen when looking down. In addition, his March MRIs showed that his tumor was greatly reduced, his MS was stable, and he had no new CNS lesions or infarctions [22,23,24,25,26,27,28,29].

## 5. Conclusions

This patient has struggled with multiple neuroimmune illnesses since childhood and as new difficulties emerge with age. Most of his maladies have a common theme, they are inflammatory, i.e., obesity, diabetes, asthma, MS, MDD, brain tumor, and the like. To control diabetes and obesity, he monitors his AM fasting glucose, which for example from January 1 to October 3, 2019, showed that his average (± standard error) serum glucose was 109.2 (± 1.3 mg/dL) and his percent glycated hemoglobin, A1C, was 5.9 (± 0.05%). Furthermore, his min–max glucose and % A1C values ranged between 71 and 188 mg/dL and 4.4% and 8.9%, respectively. These data were collected on 271 out of 275 calendar days. At present, with yoga, self-hypnosis, gym exercises, sleep, and medications (see Box 2, in Reference [1]), he manages his illnesses, weighs 231 lb (134.1 kg, BMI = 31.7), walks a mile in 20–22 min with crutches, keeps consistent routines, performs household tasks, operates his automobile, swims, scuba dives, participates in writing and editing scientific articles and reviews with former associates, and feels euthymic (BDI January 1–October 3, 2019, was between 6 and 17) [10,11,12,13,14]. His challenges are: reducing weight (to <200 lb, <91 kg, BMI <27), preventing tumor recurrence, not allowing further degradation of his retinae (by working with specialists), stopping exacerbation of asthma and MS symptoms, maintaining serum sugar levels between 75 and 120 mg/dL, and improving balance (to prevent falls) and hands coordination for small motor functions [23,24,25,27]. Important lessons he has learned to help him manage self are: to not give into overwhelm, discouragement, and anger when an existing disease is aggravated, or a new health problem arises. He understands that these will happen with age, given his background, and overall health. So, regardless the malady, he must act to quell pain, disability, and overwhelm with NSAIDs, stimulants (teas, ginko, coffee, etc.), dietary supplements, physically exhausting self in gym to sleep, and psychological methods. The patient has learned to carefully note how his prescribed medications work (i.e., absorption, distribution, metabolism, side effects, and elimination) and endures treatment hardships to heal. He enjoys working with individuals that help him discover new medications, supplements, and psychological methods to quickly emerge from illness, depression, and helplessness. He maintains routines for medications, self-hypnosis, yoga, physical exercise, mental imagery, and derives pleasure from enjoyable activities (like scuba diving). Furthermore, he academically studies symptoms and new treatments for extant and new problems, discusses them with physicians, and engages in guided autoexperiments that produce desired results. 

While we have, at present, no control over our genetic inheritances, we can momentarily accept status quo, and then manage phenotype to live rewarding lives. Finally, we want to see similar case reports from readers (scientists, physicians, and patients) so much insight is created in the literature to help chronically ill patients personalize care for their health and wellbeing to live productive rewarding lives. 

## Figures and Tables

**Figure 1 brainsci-09-00281-f001:**
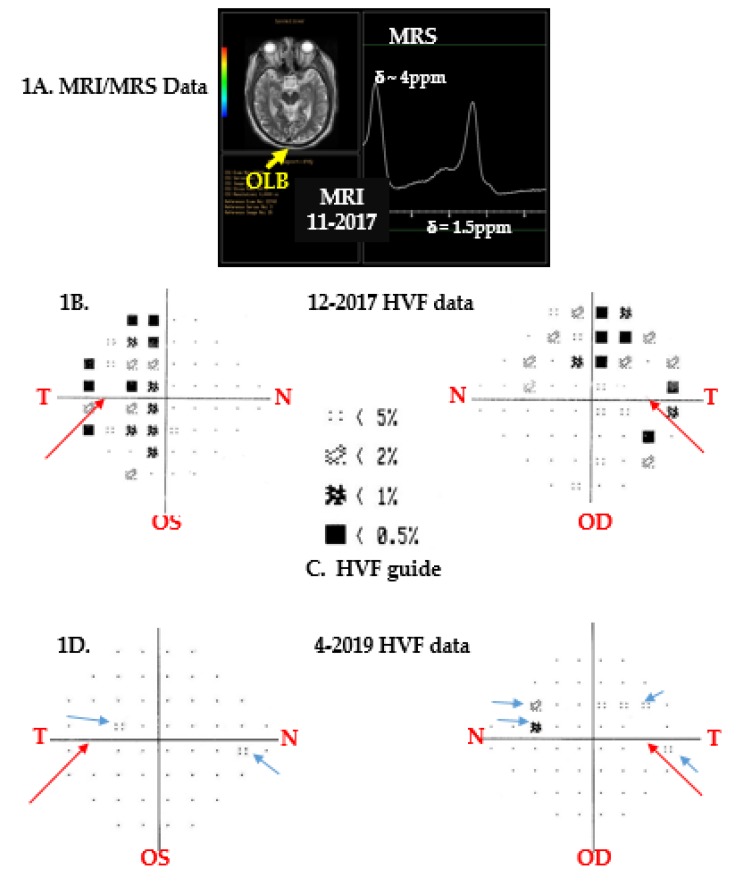
Patient’s tumor and Humphrey’s Visual Field Data.

**Figure 2 brainsci-09-00281-f002:**
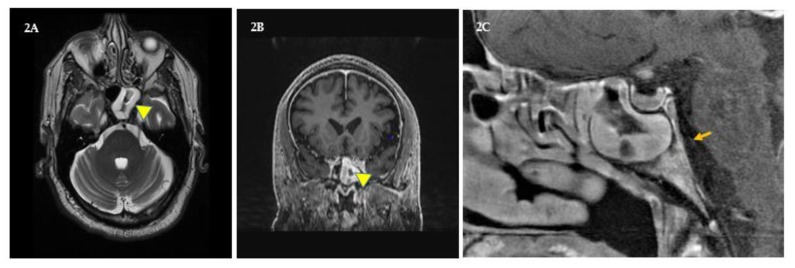
Sphenoid sinusitis. (**A** and **B**) shows transaxial and coronal views of inflammation in the patient’s left sphenoid sinus (yellow triangles). (**C**) shows a sagittal view of his congested right sphenoid sinus and the plausible path of his right sixth cranial (abducens) nerve (orange arrow) along the pons. However, congestion in the right pons cranial nerve-VI (CN6) region was not seen, and increased intracranial pressure as cause of CN6 palsy was not ruled out.

**Table 1 brainsci-09-00281-t001:** Patient’s retinal nerve fiber layer thickness (RNFLT μm) data for December 2017 and April 2019 compared with statistical information from age- and sex-matched individuals in the database.

Right Eye (OD) Dec-2017							
Sector	T	TS	TI	G	N	NS	NI
Patient	85	176	136	101	72	93	91
Statistic	72	131	138	96	72	102	104
**Overall Patient OD RNFLT**: S = 135; I = 114; T 85; and N = 72							
**Left Eye (OS) Dec-2017**							
Patient	**44**	124	126	90	82	116	99
Statistic	72	131	138	96	72	102	104
**Overall Patient OS RNFLT**: S = 120; I = 112; T = 44; N = 82							
**Right Eye (OD) Apr-2019**							
**Sector**	**T**	**TS**	**TI**	**G**	**N**	**NS**	**NI**
Patient	55	127	128	87	70	94	94
Statistic	72	131	138	96	72	102	104
**Overall Patient OD RNFLT**: S = 111; I = 111; T = 55; and N = 70							
**Left Eye (OS) Apr-2019**							
Patient	**40**	117	119	**80**	63	102	98
Statistic	72	131	138	96	72	102	104
**Overall Patient OS RNFLT**: S = 109; I = 108; T = 40; and N = 63							

Numbers in **bold red** font showed that his left RNFLTs were 39% and 55% below average for years 2017 and 2019, respectively, in comparison with age-matched Caucasian men in the database. Value in **bold blue** font showed that his left foveal (G) RNFLT in April 2019 was 16% below average, but was within normal range in December 2017 (only ~6% below average). RNFLTs in black font were comparatively within normal limits.

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
