# Peer review of "Post-Craniopharyngioma and Cranial Nerve-VI Palsy Update on a MS Patient with Major Depression and Concurrent Neuroimmune Conditions"

_brainsci, 2019, doi:10.3390/brainsci9100281_

Round 1

Reviewer 1 Report

This is an interesting and rather unique piece of clinician/author/patient case reporting. I have a few minor suggestions, mostly grammatical, and one more fundamental suggestion.

Title: the title is a little clunky and could perhaps be revised.

Line 24: MDD should be written out in full even in abstract before being abbreviated

Line 57: " and appreciate fulfilled lives" could be changed to "and hopefully improve their quality of life"

Line 61: "colorblindness" I'm not sure if this is more commonly two words or one.

Line 84: "NSAIDS" should have a small s, i.e. NSAIDs

Line 88 and 89: this sentence could be revised for the sake of grammar as it mixes imperfect and perfect tenses.

Line 109: I would argue that one can not keep one's self well by "being fulfilled", this is rather a consequence of one's activities and outlook.

Overall the title appears to be about this patient's (and author's) methods for keeping himself well despite his physical health problems, but there is a mis-match between this and the amount of the article that is actually describing his physical health problems and their impacts, rather than the methods he uses to keep well. Perhaps a little more discussion of what particular methods worked for each symptom would be helpful.

Author Response

BrainSci-591599, Sachinvala ND, et al

Disposition of reviewer’s comments, by NS and AS

Brain Sciences, MDPI’s INSTRUCTIONS:

Please provide a point-by-point response to the reviewer's comments and either enter it in the box below or upload it as a Word/PDF file. Please write down "Please see the attachment." in the box if you only upload an attachment. An example can be found here (/bundles/mdpisusy/attachments/Author/Example for author to respond reviewer - MDPI.docx?e9a1de4761e1496f).

Reviewer 1's Comments and their disposition by authors NS and AS.

Reviewer 1: The title is a little clunky and could perhaps be revised.

Authors NS and AS disposition: We shortened the title. The new title now reads:

Post Craniopharyngioma and Cranial Nerve-VI Palsy Update on a MS Patient with Major Depression and Concurrent Neuroimmune Conditions.

Reviewer 1’s Table: Requested need to improve the introduction

          Authors NS and AS disposition:  The introduction was revised.

Reviewer 1: Line 24: MDD should be written out in full even in abstract before being abbreviated

Authors NS and AS: Yes, this correction was made see attached file: Brainsci-591500 (REVIEWER and NS and AS REVISED 10-7-19)

Authors NS and AS: Please reintroduce the colon (:), and statements that follow a colon should be bullet points. In accordance with ACS’s guidelines, since each bullet point ends with a period (full-stop), the last bullet point must start with a capital And.  See MDPI’s approved ACS Style Guide, web reference: https://pubs.acs.org/userimages/ContentReviewer/1246030496632/chapter14.pdf

Lines 36-45.  The semicolons after each statement in the original text, were replaced with a period (or full-stop).

Lines 51-56 starting with “In” this report…-to-……..quality of lives" are a new paragraph

Reviewer 1: Line 57: " and appreciate fulfilled lives" could be changed to "and hopefully improve their quality of life"

Authors NS and AS: Line 56: The ending of this line should appear as:”…and improve the quality of their lives.”

Reviewer 1: Line 61: "colorblindness"

Authors NS and AS:  color blindness now appears as two words.

NS and AS: Lines 77-89.  Statements following a colon should be bulleted, and each bulleted statement should end with a period (full-stop).  The final bullet point starts with a capital And.  See ACS Style Guide referenced above.

Reviewer 1: NSAIDS should be changed to NSAIDs

Authors NS and AS Disposition: Line 83.  NSAIDs now appears with a small s as NSAIDs.

Reviewer 1: Regarding lines 88 and 89

Authors NS and AS Disposition: Respectfully, we want to keep this sentence as written.

Authors NS and AS, Line 94. The highlighted text should be replaced with: See Box 2 in reference 1.

Authors NS and AS, Line 109: Words “……being fulfilled…..” in the original text were replaced with "maintaining a positive outlook [20]."

Reviewer 1's final comments: Overall the title appears to be about this patient's (and author's) methods for keeping himself well despite his physical health problems, but there is a mis-match between this and the amount of the article that is actually describing his physical health problems and their impacts, rather than the methods he uses to keep well. Perhaps a little more discussion of what particular methods worked for each symptom would be helpful.

NS and AS Disposition: In compliance with Reviewer 1's suggestions, we corrected this perceived mismatch by changing our title and reworking our text.  For example, for the new setback, diplopia due to sphenoid sinusitis, the author’s tried and tested methods for gaining relief and becoming productive again worked. First it was important to quell the pain, observe changes in appearance and health (inward turning of right eye), maintain all other routines, and when reasonably pain free, the author studied his new problem from references on diplopia. He learned that his symptoms (see revised text) matched those of others who suffer diplopia due to CN6 palsy and was not due to CN3 or CN4 palsy. Upon meeting his neuro-ophthalmologist this speculation of CN6 palsy was confirmed by physical eye exams, and prism cover tests, and the patient was prescribed a 10diopter Fresnel plastic lens to wear over his right eye.  Furthermore, as nasal lavage turned dark and sinus pain intensified with time, he was prescribed antibiotics and then was symptom free.  A month later at the time of his brain scans for changes in his tumor, he experienced the same symptoms of pain, increased intracranial pressure, and double vision. This time in addition to his tumor’s status, his MRIs showed sphenoid sinusitis, which is known to cause double vision owing to increased intracranial pressure on CN6.  And the patient learned that his methods described in the introduction worked for him to gain relief and not feel depressed and helpless, because efforts to study and emerge from illness and become functional again for this author are powerful antidepressants.

Furthermore, our current manuscript (intended for publication in Brain Sciences) differs from our previous report in that it:

(1) updates the status of the author's brain tumor 15 months after radiation treatments were completed;

(2) explains the symptoms, diagnosis, treatment, and resolution of cranial nerve VI paralysis and diplopia; and

(3) shows how this author's methods discussed in the introduction for sustaining euthymia and productivity also helped him emerge from new health setbacks, because solving problems and emerging from illness, helplessness, depression, and being productive again for this patient are very calming.

Reviewer 2 Report

Inspiring story.

I am having a little of hard time to understand how this is different from initial publication regarding the purpose of this publication. What is the purpose of the publication?

It would be prudent to provide emphasis to differentiate this publication from the initial case report not to look repetitive.

Focus on genetic, immunologic, historical information regarding treating conditions and avoid general statements for other neurological conditions(69-70).

122-127: provide some background information and images for normal findings  vs findings of craniopharyngioma, especially given lack of biopsy. Provide the initial imaging findings of MS if available, images when diagnosed the suprasellar tumor and images after treatment. Perhaps three rows of comparative imaging.

133: provide exact numbers of normal range vs abnormal numbers and interpretation.

151:provide a specific statement regarding findings before and after treatment with specific follow up interval including percentages of improvement if available.

153-160: provide lay language to explain the findings reported and their interpretation.

184-185: perhaps a summary of the table in lay language, might make it easier to understand the purpose of the report.

186: "...for years 2016-2019". data provided for years 2017-2019.

214-218: provide comparative images at the time of diagnosis and after treatment. How the reported response is different from what is already expected from the treatment provided?

220-233: Hard to tell how the conclusions are relevant to the title.

Any follow MRIs to document changes from initial disable status due to MS to walking briskly with crutches?

Regarding a case report, it requires the scientific evidence for every reported change in his neurological status. If Imaging, Neurological Examination, Ophthalmology examination and Instrumental activities of daily living are monitored at the time of diagnosis and months or years after radiation perhaps a comparative table of all the parameters might be prudent to emphasize changes over time.

Author Response

BrainSci-591599, Sachinvala ND, et al

Disposition of reviewer’s comments, by NS and AS

Brain Sciences, MDPI’s INSTRUCTIONS:

Please provide a point-by-point response to the reviewer's comments and either enter it in the box below or upload it as a Word/PDF file. Please write down "Please see the attachment." in the box if you only upload an attachment. An example can be found here (/bundles/mdpisusy/attachments/Author/Example for author to respond reviewer - MDPI.docx?e9a1de4761e1496f).

Reviewer 2's revised review received 9-24-2019:

“Does the introduction provide sufficient background and include all relevant references? ( ) (x) ( ) ( )

NS and AS disposition:  Yes we rewrote our introduction to meet these requirements.

“Is the research design appropriate? ( ) ( ) (x = must be improved) ( )

Are the methods adequately described? ( ) ( ) (x) ( )

Are the results clearly presented? ( ) ( ) (x) ( )

Are the conclusions supported by the results? ( ) ( ) ( ) ( )

Inspiring story.”

NS and AS Disposition regarding the table on research design, methods, and results:

This revised version of our manuscript, was expanded, and we believe that it meets the requirements of Reviewers 1 and 2. 

Full new sections on Materials and Methods and Results and Discussion were introduced to explain how anatomic changes associated with tumor remission were studied by imaging and spectroscopy; and how associated vision changes were studied by Humphrey’s visual field tests; optical coherence tomography; and simultaneous cover and prism tests to reveal underlying causes, that upon resolution relieved symptoms, and felt rewarding as the patient regained some or all functions. 

Reviewer 2's Initial Comments: “I am having a [little of] hard time to understand how this is different from initial publication regarding the purpose of this publication. What is the purpose of the publication?  It would be prudent to provide emphasis to differentiate this publication from the initial case report [so as] not to look repetitive.”

NS and AS disposition: The purpose of this manuscript intended for publication in Brain Sciences, 2019, is to:

Succinctly explain with new examples that the patient discovered self-help methods discussed in NDT-2018 continue to work for him (patient and primary author), especially when his new conundrums emerged; Provide evidence for his new ailments and remedies that worked, And without being redundant, discuss changes felt by the patient after treatments were completed, i.e., relief from illness and helplessness, and joy for becoming intellectually productive again.

The NDT 2018 paper:

(a) Introduced the patient/author, and showed his multiple illnesses: [Ehlers Danlos syndrome, urticaria, chronic major depression, amblyopic left eye, overall poor vision, dyslexia, selective IgM deficiency, asthma, multiple sclerosis, and his difficulties with craniopharyngioma bitemporal vision loss].

(b) Attributed the patient’s multiple illnesses to being born in a closed Indo-Iranian community wherein examples of patients with like neuroinflammatory conditions due to consanguineous relations exist.

(c) And showed how the patient learned to manage his depression, and disability to attain and maintain euthymia.

The purpose and content of our present paper (for publication in Brain Sciences, MDPI, 2019) differ from the 2018 NDT publication because:

(a) The authors precis the patient’s background, and summarize the methods he learned to manage illnesses and disabilities to maintain euthymia and productivity. And they show that the methods he learned earlier continue to help the patient as new health challenges arise. 

(b) Then, in Figure 1 the authors show the status of his brain tumor post radiation and that it reduced in size with radiation treatments.

(c) Thereafter the authors explain concepts by Chernov and others [refs.16, 17, and 18] to show that proton MR-spectroscopy can be used to differentially diagnose craniopharyngioma in patients when surgical biopsy is not possible.  This is because only papillary craniopharyngiomas have a unique proton spectrum, not seen in any other perisellar tumor. And surgical biopsy was avoided in this patient, because of his immune problems (infections take long to resolve with antibiotics), MS, asthma, and his EDS prolongs clotting times.

(d) In Figures 1b, 1c, and 1d, the authors explained patterns of the patient’s Humphrey’s Visual Field Tests (HFT, in comparison with age and sex matched subjects), to show that presence of a large suprasellar tumor between the anterior pituitary and the anterior optic chiasm temporarily caused bitemporal vision loss, which was gradually regained when the tumor was shrunk to 0.18Ml, and although 15 months after stereotactic radiation treatments ended he regained peripheral vision, but showed central scotomas (Figure 1d). 

(e) In Table 1, the authors showed that losses in the patient’s vision due to aging, inflammatory diseases, obesity, chronic diabetes, MS, craniopharyngioma, and cranial nerve VI (abducens nerve) palsy affected his retinal and retinal nerve fiber layer thicknesses (RNFLT). These were measured using optical coherence tomography and compared with data from age and sex matched Caucasian subjects in the RNFLT database. 

(f) Finally, in their 2019 manuscript, the authors explain that despite old and new difficulties, the patient continues to use his heuristic, to be free of pain, maintain routines for physical and mental well-being, study his problems in detail, and then auto-experiment with physician to overcome adversity with new treatments, and as much as possible gain freedom for disease related helplessness, to have a happy productive life. Furthermore, he wants to read other people’s experiences in peer reviewed journals, so we all can learn new heuristics (tested methods) from each other to have improved quality lives despite being chronically ill.

Thus, the two papers are different.

Reviewer 2 lines 69-70: Focus on genetic, immunologic, historical information regarding treating conditions and avoid general statements for other neurological conditions (69-70).

NS and AS disposition:  Respectfully, we request that these lines (69-70 or thereabouts in the revised manuscript) are retained as written. This is because they disclose the patient’s state of mind to want to maintain mental lucidity, euthymia, and productivity despite numerous illnesses. And he wrote the statements, because, many physicians in US who treat people from South Asia (like the patient), have no clue how to address issues underlying patients’ concerns and illnesses from that region of the world.

Reviewer 2, lines 122-127: provide some background information and images for normal findings vs findings of craniopharyngioma, especially given lack of biopsy. Provide the initial imaging findings of MS if available, images when diagnosed the suprasellar tumor and images after treatment. Perhaps three rows of comparative imaging.

          NS and AS disposition:  The primary author can do this here for Reviewer 2 in this disposition of reviewer’s comments. And this is a side interest and not the purpose of the paper.  Furthermore, gaining such details did not improve the patient’s and his neurophysiologists (up to now 4 consulted at high quality hospitals and research centers in the US) understanding of the problem(s), so biological, chemical, physical, yoga, psychological (mind body), and other therapies could be employed to help the patient regain lost functions, enable him to work in a lab as a scientist, hold and manipulate pipettes and other equipment, play music, or find simple directions underwater with a compass and lubber line (while scuba diving), as he once did. The author says so because, as shown below, periventricular and corpus callosum (white matter just above the ventricles) lesions are present along their entire lengths in the patient’s MRIs. In addition, the patient shows cerebellar lesions as well. Periventricular and corpus callosum lesions indicate that the patient has poor R/L-coordination, balance, cognitive difficulties (poor direction finding, unable to multi-task), mood swings, and much major depression. Cerebellar lesions indicate ataxic gait (un-coordinated walking), dysarthria (slow speech), and intention tremors (hands shake when gripping and manipulating objects).  And while the patient religiously does his yoga, and physical therapy exercises to improve balance, hands coordination, and to walk smoothly without crutches each day (from 2005-present), these functions have not satisfactorily improved.  Over time, lesions fade in intensity (in MR-images, without and with contrast). However, their fading has not improved the patient’s lost somatic and cognitive functions.

Thus far his improvements are in large motor functions. He got out of bed and wheel chair, transited from various walkers to crutches, and walks briskly with them. MRI’s do not show such improvements, MRI’s show, that ring expanding lesions have subsided, and were replaced with images of damaged tissue after the attack subsided.

Furthermore to comply with Reviewer 2’s request to make a Lesions versus Functions Lost or Gained with Time table, the primary author called hospitals in New Orleans and elsewhere he once frequented for records. He was advised, that because the patient did not consistently travel to these centers after his treatments (Novantrone or Mitoxantrone, Interferon beta-1-alpha, Natalizumab, some physical therapy, etc.) ended (completed or stopped because side effects), the hospitals are not obligated to maintain his records after 5-10yrs, and cited US law for not maintaining records of inactive patients (see: https://www.healthit.gov/sites/default/files/appa7-1.pdf).  So, the table below was prepared for Reviewer 2, from personal CD’s the patient kept from his hospital stay (Feb through August 2005) at Ochsner. His 2006-2014 records do not exist. Thereafter, MRI data were not obtained until the time the patient complained of intense head aches and his brain tumor was detected (2017). 

Patients, MS lesions, from Feb-Aug 2005, were observed using a GE or a Siemens 1.5T MRI instrument

Feb-14-05, T2 Axial

May-8-05 T1 Axial

Aug-5-2005, T1 Axial

Apr-30-17 PostAxT2, 1.5T

  Sep-13-18 Post AXT2Flair, 3T

Mar-14-19 AXT2FlairW, 3T   

Glossary: Axial = Image was taken in the orthogonal plane 90 degrees rotated from the direction of the long axis of the human body.  Flair = fluid attenuated inversion recovery.  It is a MRI pulse sequence that suppresses the effects water in cerebrospinal fluid can have on the image.  Post = Image produced after a contrast enhancing reagent was introduced into the body.  T1 and T2 are MRI pulse sequences that produce different image contrasts. In images from T1 weighted pulse sequences, fat (white matter) appears bright, and in images from T2 weighted pulse sequences both fat and water appear bright.  Here T stands for time.  And in 1.5 T and 3T: Here T stands for magnetic field strength in Tesla units. 1T = 1V*s/m^2

Images and Spectra of the Patient’s Parieto-temporal Lesions in 2018 obtained using a GE 3Tesla machine

Above: In comparison to the labeled spectrum on the right. The left and central MRI/S show parieto-temporal lesions (boxed) and their spectra show abnormal intensities of proton peaks for lipids, N-acetyl-acetamide (NAA), glutamic acid and glutamine (glx), creatine (Cr), choline (Cho) and myoinositol (myo), the most abundant neurochemicals in the brain and spine.  The third image shows a labeled 1H spectrum of a normally appearing white matter (NWAM) voxel.

The figure above, shows that MS lesions at disease onset appeared large and ring expanding (see data for Feb 2005) and bright.  Ring expansion indicates spreading of disease.  In time and with treatments, his lesions contracted, their intensities diminished, and complete function improvement did not occur (remains a hope), and the patient was told that his MS is in remission.  This is because, and as the reviewer knows, lesions indicate neuron and corresponding function loss.  And CNS (brain and spine) neurons once lost are very rarely repaired to restore full function. Medications, yoga, physical therapy, etc., help neural plasticity, and the patient may recover some functions.  Neural plasticity, as the reviewer knows, is the ability of uninjured healthy neurons near the injured neurons in the brain and spine to take on new functions throughout life. Such recovery might be helped for some patients, with efforts to impose will on self to want to change status quo.  E.g., in this patients case, it was to get out of bed and wheel chair, then transit to walkers, and later crutches with thoughts (seeing patient’s efforts resulting in improvements in his mind’s eye) and daily performing hours of physical exercises (see reference 1).

Reviewer 2 line 133: provide exact numbers of normal range vs abnormal numbers and interpretation.

Authors’ NS and AS disposition:  Respectfully, the exact patterns and numbers are in Figure 1 and Table 1.

Reviewer 2 line 151: provide a specific statement regarding findings before and after treatment with specific follow up interval including percentages of improvement if available

Authors’ NS and AS disposition: This is provided in the revised Table 1 and clarified in the revised text.  We thank Reviewer 2 for pointing this out.

Reviewer 2, line 153-160: provide lay language to explain the findings reported and their interpretation

Authors’ NS and AS disposition: Lines 153-160 in the earlier version of our paper discussed the legend to Figure 1. Figure 1a shows the tumor (boxed structure), its proton spectrum, and states that despite reduction in size from 3mL before radiation treatments, to 0.18mL after treatments, the proton spectrum of the tumor remained unchanged, and showed the same alkyl and hydroxyl compound resonances at (~1.5 and ~4) ppm respectively, indicating the presence of fatty acids and cholesterol, and hydroxyl compounds like water and inositol in the soft tumor. Furthermore, the legend to Figure 1a alluded to occipital lobe bending (OLB).  OLB is a sign, seen in MRIs of chronically ill individuals (see Figure 1a). It means that the occipital lobe from one hemisphere (side) of the brain, extends beyond the length of the same lobe on the other side, and appears to wrap around it.  Likewise, ventricular enlargement (self-explanatory) is also typically seen in MRIs of the brains of chronically ill patients.

Figure 1B, shows two visual field maps for the left (OS), and right (OD) eyes.  The visual field of each eye is the extent to which that eye can see peripheral objects while looking straight. So, when looking straight normally functioning eyes can see objects at 50degrees above the line of gaze (superiorly), at fifty degrees in the nasal direction (nasally), 60degrees below the line of gaze (inferiorly), and at 90degrees side to side (temporally).  Because the patient’s tumor above the pituitary is pressing against the optic chiasm (region of the brain where the right and left optic nerves cross sides) from below, the patient’s side to side vision is restricted as shown in Figure 1B. And loss of side to side vision loss is called bitemporal vision loss.

Figure 1C shows 4 different patterns that indicate the extent to which males of the same age at the patient, have sensitivity to light at that point on the retina. Since the greater the light sensitivity, the better the vision, the Humphrey’s visual field (HVF) tests evaluate the patient’s sensitivity to light at 54points on the retina in comparison with Caucasian males his age.  So, the four-dot patterns, when present in the patient’s HVF map, indicate that in that region of the retina only 5% of males his age show poorer vision, and 95% have better vision than the patient.  Likewise, the dark and completely blackened rectangles, when present in the patient’s map, show 99.5% of males his age in the database have better vision than the patient, and only 0.5% show poorer vision.

          Data for Figure 1D were recorded 15months after radiation treatments ended, and the patterns show that the patient’s vision had improved, bitemporal vision loss could not be seen, however, the 4-dot patterns were mostly lined up along the X-axis, indicating that the patient could be losing central vision, and this shows that the patient is showing signs of central scotomas.  Small regions along the central meridian of the patient’s retina are losing sensitivity to light. 

          In HVF tests compare the intensity = luminosity of the light (stimulus) with the patient’s sensitivity to that stimulus.  It should be noted that greater the intensity of light needed to see an object, the poorer is that individual’s sensitivity to the stimulus.  And conversely the lesser the intensity (luminance) of light needed to see an object (stimulus) the greater is the person’s sensitivity to light.  Let us call the unit of brightness or luminance, Apostilbs (asb) and let us call the units of sensitivity decibels (dB). Incidentally, the word Apostilb comes in physics from the religious term “apostle,” “the enlightened one” in Greek. Since the reviewer wants her or his information at the level of a lay person’s understanding, let us only consider the underlined axiom above (interpreted from the reference at the end of this paragraph). If luminosity is 1000 brightness is high, and individual’s sensitivity is less and approximately equal to 10.  At 100abs luminosity brightness is 10times less, and correspondingly, sensitivity is 20decibels = 20dB.  So at light intensity of 10 asb, sensitivity = 30 dB (which is equal to the lighting (stimulus and background) conditions used in HVF-tests.   https://www.haag-streit.com/fileadmin/Haag-Streit_Diagnostics/perimetry/Visual_Field_Digest/Chapter_2/Visual-Field-Digest_chapter-2_01.pdf

Reviewer 2, lines 184-185: perhaps a summary of the table in lay language, might make it easier to understand the purpose of the report

Authors AS and NS disposition: Table 1, compares the patient’s retinal fiber layer thicknesses measured by computed laser tomography.  The text describing the table and legend below it is very much in line with such summaries (of the RNFLT-test and results), as they appear in professional journals.

And respectfully, since reviewer 2’s demands an explanation in lay language (not found in professional journals), the primary author will comply with his her request this disposition of reviewer’s comments (and again, respectfully, we cannot write like this in professional journals). So, here we go with as lay an explanation the primary author can provide: 

As the reviewer knows, computed tomographic images can be cross-sectional representations (or slices) mathematically cut from a three dimensional model of an object from top (anterior) to bottom (posterior) 90degrees to the long axis, and each slice is called a cross-sectional slice).  If the object is cut from front (anterior or ventral) to back (posterior) along the long axis, then a slice is called a coronal slice. And a slice from side to side slices, is called a sagittal slice.  For X-ray tomography the cutting tool is an X-ray, for MRI the cutting tool is radio waves in a magnetic field and mathematically defined volume elements.

For these optical measurement, the cutting tool is 870nm laser light, and thicknesses are measured in microns.  Furthermore, one micron (mm) is one millionth of a meter (1mm = 1x10^6m).  In this cross-sectional tomographic evaluation of the patient’s eye, the cross-sectional slice was cut at the level of the ciliary body, the fovea, the macula, and the optic cup was studied, to expose thickness of the layer of fibers (axons) that enter the eye at the optic cup (blind spot) to go to the brain.  Thicknesses of the retinal nerve fiber layer (RNFL) were measured near the temple (T), the globe (G or fovea, which is the region of the retina with the highest number of cones and rods denoting highest sensitivity to light, and the region of the retina near the nose, (N).  Additional RNFL thicknesses are measured above and below in the temporal region and are designated as temporal superior (TS) and temporal inferior (TI). Likewise, thicknesses are measured in the nasal superior and inferior regions, designated NS and NI.  All thickness values are then compared with like values of age and sex matched individuals in the RNFLT data base of Heidelberg Engineering, Inc.’s instrument.   

Reviewer 2, line 186: "...for years 2016-2019". data provided for years 2017-2019.

Authors NS and AS disposition: Yes we corrected this error.

Reviewer 2, lines 214-218: provide comparative images at the time of diagnosis and after treatment. How the reported response is different from what is already expected from the treatment provided?

Authors NS and AS disposition.  The description for Figure 2 has been updated in the revised text.

Reviewer 2, lines 220-233: Hard to tell how the conclusions are relevant to the title. Authors NS and AS disposition: Thanks, yes, we changed the title and the text.

Reviewer 2: Any follow MRIs to document changes from initial disable status due to MS to walking briskly with crutches? Authors NS and AS disposition: As mentioned earlier, I only have MRIs for 2005 and then from 2015 when the tumor was detected.

Reviewer 2: Regarding a case report, it requires the scientific evidence for every reported change in his neurological status. If Imaging, Neurological Examination, Ophthalmology examination and Instrumental activities of daily living are monitored at the time of diagnosis and months or years after radiation perhaps a comparative table of all the parameters might be prudent to emphasize changes over time.

Authors NS and AS disposition:  Doing justice to this request, could likely become several PhD theses and not a case report.  Please go to pubmed.gov, and type “examples of single patient case reports.  The reviewer will find about 200 titles of which 50 articles are free and open access.  Not a single report we studied has the kind of case report details Reviewer 2 is seeking.  None the less, it is a good request.  Thank you for taking the time to afford us a very challenging and enjoyable review.

Reviewer 3 Report

The content of your paper is from a very unique perspective.  As scientists and clinicians, we often overlook the psychological impact that a diagnosis has on a patient once he/she leaves the office.  This paper is an important learning tool for all of us - both highlighting the science behind several auto-immune diseases and elucidating the psychological impacts/depression treatments as they affect a patient over time.  

Author Response

BrainSci-591599, Sachinvala ND, et al

Disposition of reviewer’s comments, by NS and AS

Brain Sciences, MDPI’s INSTRUCTIONS:

Please provide a point-by-point response to the reviewer's comments and either enter it in the box below or upload it as a Word/PDF file. Please write down "Please see the attachment." in the box if you only upload an attachment. An example can be found here (/bundles/mdpisusy/attachments/Author/Example for author to respond reviewer - MDPI.docx?e9a1de4761e1496f).

Reviewer 3’ comments: The content of your paper is from a very unique perspective. As scientists and clinicians, we often overlook the psychological impact that a diagnosis has on a patient once he/she leaves the office. This paper is an important learning tool for all of us - both highlighting the science behind several auto-immune diseases and elucidating the psychological impacts/depression treatments as they affect a patient over time.

Authors NS and AS disposition:  Thank you for your generous words of praise.

Reviewer 4 Report

The early heat sensitivity and later, tumor related sensitivity should be better described 

There should be a physician reference to support the claim that the tumor is actually pressing against in Figure 2c

Author Response

BrainSci-591599, Sachinvala ND, et al

Disposition of reviewer’s comments, by NS and AS

Brain Sciences, MDPI’s INSTRUCTIONS:

Please provide a point-by-point response to the reviewer's comments and either enter it in the box below or upload it as a Word/PDF file. Please write down "Please see the attachment." in the box if you only upload an attachment. An example can be found here (/bundles/mdpisusy/attachments/Author/Example for author to respond reviewer - MDPI.docx?e9a1de4761e1496f).

Reviewer 4’s Comments: The early heat sensitivity and later, tumor related sensitivity should be better described There should be a physician reference to support the claim that the tumor is actually

pressing against in Figure 2c. 

Authors AS and NS disposition to Reviewer 4’s comments: All MS patients are unable to deal with high day time summer temperatures in Southern US States.  These typically are more than 95deg-F with >90% relative humidity.  During early years with multiple sclerosis, and with a brain tumor, even now after remission, the author feels weak stepping out of the car in mid-day sun, later he suddenly loses balance and falls.  His blood pressure is low to near normal, he does not sweat, and it feels like a heat stroke.  Perhaps this reference might help reviewer 4:  Bol Y, et al. Fatigue and heat sensitivity in patients with multiple sclerosis. Acta Neurol Scand. 2012 Dec;126(6):384-9. Incidentally, after this patient started using S-Adenosyl-L-methionine ~400mg daily, he can walk for about 15-20 min in the summers, at about 2 to 4 PM when it feels very hot and muggy.

In Figure 2c, CN6 does not appear to touch any structure in the lower pons region, the increased intracranial pressure with sphenoid sinusitis is proposed to cause headaches, dizziness, and diplopia.  Thank you for catching this for us, and we changed our text.